# Evolution of Global Health and Psychosocial Factors among Hospital Workers during First Year of SARS-CoV-2 Pandemic: A Longitudinal Study

**DOI:** 10.3390/ijerph192215260

**Published:** 2022-11-18

**Authors:** David Lucas, Sandrine Brient, Tanguy Le Grand, Jean-Dominique Dewitte, Brice Loddé, Richard Pougnet, Bisi Moriamo Eveillard

**Affiliations:** 1ORPHY Laboratory, University Brest, F-29200 Brest, France; 2Occupational Health Service, Teaching Hospital, F-29200 Brest, France; 3Laboratoire d’Etude et de Recherche en Sociologie (EA 3149), Université de Brest-Bretagne Occidentale, F-29200 Brest, France

**Keywords:** occupational health, SARS-CoV-2, healthcare workers, mental health

## Abstract

Objectives: To assess trends in overall health (mental and physical) and psychosocial factors in a population of workers (both healthcare and non-healthcare) in a French teaching hospital during the first year of the SARS-CoV-2 pandemic in France. Methods: A validated version of the SATIN questionnaire with adapted scoring was used to collect data on health and impacts of work stressors. This questionnaire was sent to all workers at the hospital in T1 (July–August 2020) and T2 (July–August 2021) and self-administered online. Results: A total of 1313 participants who completed the questionnaire at T1 and 826 at T2 were included. Overall, 568 workers completed the questionnaire at T1 and T2. We found a deterioration in overall health and especially stress and mental health in hospital workers and healthcare workers (HCWs), with a negative impact of the workload and work environment. Conclusions: The SARS-CoV-2 pandemic impacted negatively the mental health, work stressors, and psychosocial perceptions of both HCW and non-HCW in a French hospital. The study confirms that hospital workers are an important target.

## 1. Introduction

Since the beginning of 2020, the SARS-CoV-2 pandemic has had a significant strengthening effect on healthcare systems, including healthcare workers (HCWs).

Studies on HCWs’ well-being and mental health have, however, regularly reported problems. These include symptoms of post-traumatic stress, burnout, depression, and anxiety, associated with their occupational activities both during epidemics and at other times [1,2,3,4].

A large number of studies on the mental health impacts of the SARS-CoV-2 pandemic on HCWs were firstly published in China [5,6,7,8]. The mental health impact of the SARS-CoV-2 pandemic on HCWs has emerged as a global concern. In systematic reviews and meta-analyses, authors have found a high prevalence of anxiety, depression, acute stress, post-traumatic stress, and sleep problems, with sleep disorders increasing over time [9,10,11,12]. In a review focusing on Asian HCWs, the risk factors found for mental health disorders included being female, being a frontline worker, being at the age extremes of the workforce (youngest and oldest HCWs), being a nurse, having direct contact with infected patients, having worked for fewer years, having longer working hours, or there being a lack of individual protective equipment. On the contrary, psychological resilience and family or a committed relationship were identified as protective factors [13]. Such data have also been reported in other parts of the world [6,14]. We also found that being a medical resident leads to a risk of stress, mental health impacts, and a negative perception of the organizational context and work demands [15].

Few studies provide data on physical symptoms and/or psychosocial factors, such as organization, work activity, or activity management, for HCWs over this period. In a narrative review of studies conducted in Singapore compared with Chinese data, Ng et al. found a lower prevalence of psychological symptoms but a higher prevalence of physical symptoms, which, they concluded, were due to somatization [16].

For other hospital workers not involved in healthcare, few studies have been published on the SARS-CoV-2 pandemic’s impact on their mental and physical health [1]. Most of them are Chinese and performed during the first outbreak. Compared to administrative teams, health professionals were 1.4 times more prone to experiencing fear and suffering from anxiety and depression [2], and medical teams, especially those in Wuhan, had a higher level of stress than university students [3]. Medical health workers showed higher prevalence rates of insomnia, anxiety, depression, somatization, and obsessive-compulsive symptoms than non-medical health workers [4]. In a longitudinal study, Sasaki et al. noted, after adjusting for covariates, that psychological distress (and subscales of fatigue, anxiety, and depression) as well as fear and worry about COVID-19 increased statistically significantly more among healthcare than non-healthcare workers from period 1 to period 2.

In a previous study, we also found that being male and a HCW were risk factors for having a negative perception of work demands and the work activity environment [17].

Within France, HCWs experienced a strong impact on their mental health during the first wave of the SARS-CoV-2 pandemic, with a stronger deterioration during the following waves. Psychosocial factor and working environment (physical and mental) assessments underlined the main occupational factors of the individual and collective working environment. In this field, SATIN is a transversal questionnaire developed for preventive medicine targeting well-being at work by the National Institute of Research on Security (INRS) in France [18,19]. Using this validated questionnaire, we performed a study on health and psychosocial factors in French hospital workers at the end of the first wave of the SARS-CoV-2 pandemic.

Indeed, a recent narrative review points out the lack of studies and discussion on mechanisms that guarantee intervention and the sustainability of them in the literature on HCW mental health support programs. For us, the lack of attention to sustainability may also be linked to focusing firstly on disaster mental health rather than occupational health [20].

Thus, it appears to us that better knowledge of the physical and mental health impacts on hospital workers and occupational psychosocial factors with a higher impact is clearly needed and would be relevant to enhance prevention programs.

The aim of the present study is to assess trends in overall health (mental and physical) and psychosocial factors in a population of workers (both healthcare and non-healthcare) in a French teaching hospital during the first year of the SARS-CoV-2 pandemic in France.

## 2. Materials and Methods

This study is a longitudinal follow-up of French teaching hospital employees.

### 2.1. Participants

In 2020, during a hygiene and security committee, hospital trade unions, managers, and occupational health services decided to assess hospital workers’ global health. The occupational health service proposed a project that was accepted by all members of the hygiene and security committee. We assessed an online cross-sectional study of all professionals at the hospital [15,17]. The survey was conducted between 1 July 2020 and 31 August 2020 and involved all workers employed in a teaching hospital in France. The total number of workers was 7299, including 799 medical residents.

The inclusion criteria were as follows: working in the hospital for more than one year, adults over than 18 years, and willingness to join the study.

A total of 1657 workers answered the questionnaire at Time 1 (T1).

In 2021, through an e-mail from the hygiene and security committee, we further invited all workers to participate in the second step of the study. The inclusion period was 1 July to 10 August 2021 (Time 2 (T2)). The study was a questionnaire-based study, using the same questionnaire in T1 and T2. In this second questionnaire, a specific question on participation in the first step (T1) (yes/no) of our study was added. Participants were classified into three categories: P tot T1 who only answered the questionnaire at T1; P tot T2, who only answered the questionnaire at T2, and P tot T1 and T2, who responded ‘yes’ to the specific question on first step T1 in the second questionnaire at T2.

We also divided each population according to workers’ status: HCW or non-HCW status.

For more clarity, we decided to compare results from workers who completed the second assessment and those who completed the first assessment.

P HCW T1 included participants who answered only the first questionnaire (T1) and were included in the HCW group.

Medical residents, doctors, nurses, medical students, nursing assistants, midwives, paramedics, physiotherapists, and radiographers were included in the subgroup “healthcare workers” (HCWs), while other participants were included in the subgroup “non-healthcare workers” (non-HCWs). Administrative workers, laboratory technicians, maintenance staff, and cook staff workers were included in the non-HCW subgroup. None of the non-HCWs were frontline workers. The study was approved by the ethics committee of Brest Hospital, N° B2021CE19. The survey was anonymous, and confidentiality of information was assured.

### 2.2. Questionnaire

The SATIN questionnaire was developed by the University of Lorraine and the INRS [17]. Based on the SATIN questionnaire, after authors’ acceptance, we added some questions on workplace and demographic data. These added questions were not included in the SATIN questionnaire results analysis but for demographic and work outcomes. Firstly, participants had to answer specific questions on inclusion criteria and acceptance in order to be included in this survey. The last question was an open one for any remarks that workers wished to add (personal, professional, about the study).

The questionnaire was built including 86 questions in 6 parts: (1) personnel and professional identification (10 questions); (2) health reports (16 questions); (3) work strain and abilities (8 questions); (4) working environment (39 questions); (5) work assessment (4 questions); (6) supplementary questions for occupational physicians (9 questions).

Each question had five possible answers and each answer was linked to a specific score. Means of scores were calculated in each part of the questionnaire: health reports (physical health (self-evaluation of health and compared to next year), mental health (self-evaluation of mental health, confidence in the future), physical symptoms (musculoskeletal disorders), psychosomatic symptoms (headache, sleep problems, gastrointestinal problems), and stress (feeling stressed, exhausted at work, breakdown because of job)); work strain (physical, emotional, concentration, knowledge); work abilities (physical, emotional, concentration, knowledge); working environment (physical environment, work activity (interest, variety, utility, responsibility, diversity, quality of social relations), framework of activities (clarity, consistency, latitude, support, interruptions), and organizational context (number of hours, financial support, salary communication, job security, job career)); and self-assessment of work conditions in their entirety.

For each question, workers had to choose only one in five possible answers and each answer was linked to a specific score. The scores obtained were continuous scores that could vary theoretically between 1 and 5 points; scores close to 1 indicate very poor health, while scores close to 5 indicate very good health on the dimension in question. Means of scores were calculated in each part of the questionnaire: health reports (physical health, mental health, physical symptoms, psycho-somatic symptoms, and stress), work strain and ability, working environment (physical environment, activity, framework of activities and organizational context), self-assessed work conditions in their entirety. The scores of each part were interpreted as follows: <2.5, poor health or negative perception; 2.5–3.5, mild health or perception; >3.5, good health or positive perception. Global health and general workplace environment self-evaluations were scored for a second time with, respectively, health reports and working environment scores.

In the first step of this study, in 2020, we divided the demographic outcomes to perform multivariate analysis. A multinomial logistic regression analysis was performed and the associations between risk factors and outcomes were obtained after adjustment for cofounders including, gender, age, and years at workplace. Demographic data were self-reported by the participants, including occupation, gender (male/female), age (<35, 35–44, 45–54, >55 years), and time at workplace (<5, 6–15, 16–26, >26 years).

Referring to this first publication and to allow better reliability between the results of the two phases, we decide to retain such classification by categories.

### 2.3. Statistical Analyses

Data analysis was performed using R-4.0.2 and R Studio software (R Foundation for Statistical Computing, Vienna, Austria). The results for continuous variables are shown as means. The ranked data, which were ranked from each part of the questionnaire, are presented as frequencies and percentages. The mean change in scores of psychological distress and physical symptoms from T1 to T2 were compared between all workers and in the subgroup of HCWs (t-test for two independent groups). The Chi 2 test was used for the comparison of demographic data between the two populations. The significance level was set at *p* = 0.05.

## 3. Results

A total of 1657 participants completed the questionnaire at T1 and 830 at T2. Moreover, 568 workers completed the questionnaire at T1 and T2. In the subgroup of HCWs, for T1, T2, and T1-T2, respectively, 871, 530, and 211 questionnaires were included. Due to the time at the hospital being less than one year, we excluded 344 workers in the first step and 4 in the second step.

Most of the HCWs were nurses (28 and 36%), nursing assistants (12 and 7%), and medical doctors (11 and 12%) for the two periods. Most of the included workers were female (80 and 79%) and HCWs (66 and 64%). In the non-HCW group, administrative assistants (10 and 11%), laboratory technicians (3%), and technicians (2 and 3%) were the most represented. Most workers were aged between 35 and 44 years old. Demographics and occupational characteristics of the population are described in Table 1.

When looking at differences in characteristics between the two periods’ groups (T1 and T2), we did not find significant differences in gender and HCW status (Table 2).

When focusing on the HCW subgroup, differences in gender were not statistically significant (Table 3).

For results of different parts of the SATIN questionnaire, when comparing the mean scores of the workers who answered the questionnaire at T1 and those who answered at T2, we found a significant deterioration in global health (*p* = 0.01) and mental health (*p* = 0.03), as well as physical health (*p* = 0.02). Workers reported higher negative perceptions of global work assessment (*p* < 0.0001), global work environment (*p* < 0.0001), and workload (*p* = 0.03) and also more psychosomatic symptoms (*p* = 0.02) or self-evaluated stress (*p* = 0.004).

In the subgroup of HCWs, mean scores of global work assessment (*p* < 0.0001), global work environment (*p* < 0.0001), global health (*p* = 0.007), stress (*p* = 0.01), physical health (*p* = 0.01), and psychosomatic symptoms (*p* = 0.01) significantly decreased between T1 and T2 (Table 4).

## 4. Discussion

The first year of the SARS-CoV-2 pandemic had an impact on the global and mental health of French hospital workers and HCWs. We also noticed higher experiences of stress and psychosomatic symptoms in our included population. Global work assessment and global work environment are the psychosocial factors with the most significant deterioration over one year in hospital workers, including HCWs and non-HCWs. In contrast, no impact has been found for the physical environment. For other psychosocial factors, such as as abilities, organizational context, and work demands, we found a decrease in self-assessment but this was not significant. The health impact is the same in HCWs, also with a deterioration that is not statistically significant, for perceptions of workload, abilities, and organizational context.

The mental health impacts of several waves of the pandemic on hospital workers lead to the risk of mental health disorder development. Accessibility to psychological support and mental health treatment for HCWs are two key means of prevention of long-term psychological distress. In the first part of our survey (T1), our hospital offered two means for staff to contact a psychologist, one involving the phone (available 7 days a week) and the other involving the occupational health service during the week. The majority of consultations were held in the occupational health service (89%). Occupational health professionals noted anxiety disorders during the lockdown period and shortly afterwards, with an increased level of suicidal ideation and a resurgence in latent conflicts between units or employees. On the one hand, the most impacted employees were young doctors, medical unit managers, and, interestingly, those workers who had stayed at home due to higher health susceptibility to the virus. On the other hand, they reported positive impacts in some units, with better collective support and an improved sense of purpose at work [17].

The effectiveness and acceptance of psychological support are better when including basic, professional, and personal needs in association with active dialogue between HCWs and leadership. The role of leadership has been noted several times; surveys during the SARS-CoV-2 pandemic revealed that HCWs required assurance from all levels to ensure a sense of cohesion, trust building, and respect across the organization. In a narrative review on COVID-19 pandemic support programs for HCWs, based on pre-existing definitions of mental health and occupational health guidelines proposed by Australian researchers, David et al. developed different aspects of mental health prevention interventions. Another aspect of support programs is promoting access to mental healthcare. The authors highlighted the importance of promoting positivity among workers through town halls, regular feedback sessions, workshops, and peer-support interventions [20]. Depersonalization was also lowered due to satisfaction with the usefulness and timing of workplace communications in frontline HCWs [21]

In our study, French hospital workers negatively assessed their work environment in 2020 and significantly more negatively one year later. The relevance of collective support from the occupational environment was evaluated and discussed in different studies. The SARS-CoV-2 pandemic has brought about changes in the framework of the work of HCWs, which seem to be acting both positively and negatively on their psychological equilibrium. Some authors found that working on the frontline, or being a physician, positively impacts compassion satisfaction, which may have balanced compassion fatigue in this particular situation [22]. In Australian frontline HCWs, the perception of workplace support is significantly associated with a reduction in mental health outcomes and emotional exhaustion in terms of the subdomain of burnout [23]. In the first part of our study (T1), occupational health professionals reported similar conclusions, where, on the one hand, the most impacted employees were young doctors and medical unit managers, and, on the other hand, there were positive impacts in some units, with better collective support and an improved sense of purpose at work [17].

In another scoping review of guideline recommendations to reduce mental health burdens in HCWs during COVID-19, such a need for mental health support is emphasized in most of the articles included. Various strategies for individual care are described, including diaphragmatic breathing, meditation, a positive mindset, mindfulness, and relaxation. The authors stressed the pivotal role of appropriate appreciation and professional validation. Indeed, they highlighted the role of support both at the individual and at the organizational level. For communication, high-quality and transparent communication with regular updates to all HCWs is essential [24].

Different examples, including psychoeducational training with the aim to decrease psychological impacts and improve coping skills, were described. Some specific techniques, such as online or mobile applications and online programs focused on teaching HCWs about effective coping techniques, have been tested in different countries. The objectives of these new approaches are to raise the effectiveness of prevention programs by limiting obstacles as well as providing alternative programs [20].

Work and personal environments may have triggered the process and appearance of the effect on mental health, as shown by the trajectory of burnout syndrome (BOS). In our study, we only assessed the work environment impact, and we found a high mental health burden and decreased global health, global work assessment, and global work environment in the entire population of French hospital workers and also HCWs. In this field, results from a study performed during the first wave on HCWs showed that the absence of PPE was related to BOS and higher depersonalization aspects declared by participants. On the subscale of emotional exhaustion, physicians and nurses presented higher levels [7] than others. For nurses, in a meta-analysis of BOS, the prevalence of emotional exhaustion was 34.1%, that of depersonalization was 12.6%, and the lack of personal accomplishment was 15.2%. Increased workload and decreased self-confidence in caring for infected patients were found to be risk factors [21]. In Pappa’s study, anxiety and depression were more prevalent among nurses compared to physicians. One hypothesis is that they have higher workloads and different responsibilities, with direct contact with patients and decreased time for patient care [7]. In our study, we found a significant deterioration in workload for the total included population but not for HCWs. Moreover, we found a deterioration, albeit not significant, in abilities and organizational context for hospital workers and in abilities and workload for HCWs. We did not find any differences in work demands between the two periods for the total population and the subgroup of HCWs. It seems that psychosocial factors including workload, organizational context, and abilities were more strongly impacted by the pandemic. We did not compare psychosocial factors in different occupational groups in HCWs, but it could explain the absence of variation for others.

Another axis of prevention includes the provision of adapted personal protective equipment (PPE) and supplies.

In this field, results from a study performed during the first wave on HCWs showed that the absence of PPE was related to BOS and higher depersonalization aspects declared by participants. The impact of a lack of PPE has been discussed in several studies on the mental health impact of the SARS-CoV-2 pandemic [6,7,8].

In a longitudinal cohort, Sasaki et al. noticed that the degree of fatigue, anxiety, and depression, as well as fear and worry about COVID-19, increased significantly more among healthcare than non-healthcare workers from T1 (July–August 2020) to T2 (July–August 2021) [25]. As in our study, the question on psychosocial factors revealed that, for HCWs, there was both a high impact on and degradation in organizational context and global work assessment. Impacts on mental health with acute and chronic problems were linked to levels of contamination during the pandemic and the short periodicity between pandemic peaks, with increased pressure on healthcare systems. HCWs have a professional level of ability to cope with the care of patients. However, with frequent modifications to treatment and infection prevention protocols, knowledge about SARS-CoV-2, and public health policies, their adaption abilities were overloaded. For HCWs, the psychosocial factor with higher variation during this period was the organizational context, which impacted work assessment and subsequent mental and global health.

Interestingly, we found, in the total population of French hospital workers but not in the HCW subgroup, increased negative perceptions of workload at T2. Workloads were highly elevated during this period, but we can hypothesize that there was less variation for HCWs than in the total population of workers. HCWs are essential workers with a lesser impact on their professional activities during periods of lockdown. The nature of the pandemic corresponded to the primary values of caregivers, with a focus on care. Frontline units and medical care units had support from other colleagues in different units, such as surgeries, where activities were rapidly stopped. Higher feelings of uncertainty regarding the work and personal environment for non-HCWs could also be an explanation.

The deterioration of physical health and higher psychosomatic symptoms expressed by HCWs and non-HCWs, without such a negative impact over one year regarding the occupational physical environment assessment, could be linked to the mental health impacts and workload perceptions in this population. Work strain during the first year of the SARS-CoV-2 pandemic in a population of hospital workers had an impact on mental health but also on physical health, with a higher risk of musculoskeletal disorders and sick leave.

In terms of the general population, a study assessed the prevalence of serious psychological distress in the US population aged 20 and over, which was followed prospectively from February 2019 (T1), with subsequent interviews in May 2020 (T2) and August 2020 (T3). Results showed that psychological distress was associated with age, race/ethnicity, and household income at the first stage. Interestingly, the presence of serious psychological distress at Time 1, a low household income at T1, disruption of employment at T2, and disruption of healthcare at T2 were all significantly associated with an elevated risk for SPD at T3 [26].

Indeed, guideline recommendations to reduce the mental health burden in HCWs during the pandemic were analyzed in a recent paper. They classified guidelines and recommendations into four categories, namely social/structural support, the work environment, communication/information, and mental health support [21].

The participants of our study represent a large population of HCWs and non-HCWs from the same hospital. Thus, we hope that the risk of demographic bias, such as lifestyle factors and the incidence of COVID-19 infection during the study period, was reduced. We also used one instance of the questionnaire at T1 and validated it for a second time at T2. This questionnaire, SATIN, was intended for use in large organizations such as our hospital.

Our study has some limitations. Bias may have arisen from the survey being conducted via the internet. Self-reported data and the long time required to complete the survey (more than 15 min) could impact the results. The sample itself may be biased in terms of working conditions and workplaces, especially as there were very few hospital cleaners or ambulance service personnel among the respondents. All participants were from the same province, which also limits the potential for the generalization of our findings to a greater area. We did not have data on lifestyle factors, family status, or the time for which the respondents had occupied their present work positions. Lastly, the two samples, at T1 and T2, differed in their demographic data.

## 5. Conclusions

We can hypothesize that the main impact of the SARS-CoV-2 pandemic is on mental health and is specifically due to workload deterioration for HCWs. The occupational organization and workload in the hospital, and especially in care units, seem to be the most relevant axes of prevention regarding the mental health impact. In occupational and public health interventions and priorities, an emphasis should be placed on the psychological burden on HCWs. Actions promoting collective support, protective equipment, and the organizational context and framework are needed, particularly in the context of the pandemic. Developing interventions to reduce the risk of adverse mental health outcomes is imperative. The potential consequences of not ensuring a fair income for HCWs include such factors as high workloads and longer working hours, with a risk of imbalance between work and personal life, and this can cause further physical and mental strain. Indeed, attention should be directed towards long-term HCWs in mental health interventions.

## Figures and Tables

**Table 1 ijerph-19-15260-t001:** Demographics and occupational characteristics of population.

		T1		T2	
Occupation		Number	%	Number	%
	Nurse	368	28	301	36
	Nursing assitant	160	12	54	7
	Administrative assistant	127	10	91	11
	Doctor	138	11	96	12
	Medical student	81	6	12	1
	Health supervisor	78	6	51	6
	Radiographer	61	5	23	3
	Laboratory technician	40	3	24	3
	Health manager	26	2	19	2
	Technician	25	2	24	3
	Physiotherapist	21	2	13	2
	Midwife	15	1	14	2
	Other	173	12	101	12
Age in years	<25	26	2	10	2
	25–34	344	26	152	30
	35–44	408	31	304	33
	45–54	362	28	258	25
	>55	173	13	103	10
Time at workplace	1–5	334	25	154	26
	6–15	516	39	325	39
	16–25	319	24	243	25
	>26	144	11	104	10
Work status	Non-healthcare	442	34	296	36
	Healthcare	871	66	530	64
Gender	Female	1049	80	681	79
	Male	264	20	145	21

**Table 2 ijerph-19-15260-t002:** Comparison between populations included at T1 and T2.

		P tot T1 N	P tot T2 N	*p*
Gender	Female	1049	682	
	Male	264	185	0.14
	<25	39	19	0.23
Age in years	25–34	396	260	<0.001
	35–44	426	285	0.001
	45–54	370	214	0.07
	>55	174	85	0.6
Time at workplace	1–5	334	227	0
	6–15	516	342	1
	16–25	319	214	0
	>26	144	84	0.3
	HCW	871	530	
HC status	Non HCW	442	336	0.3

P tot T1: Participants included at T1. P tot T2: Participants included at T2. *p*: *p* value. HC status: Healthcare status. HCW: Healthcare worker. Non-HCW: Non-healthcare worker.

**Table 3 ijerph-19-15260-t003:** Comparisons between T1 and T2 for HCWs.

		HCW T1 N and (%)	HCW T2 N and (%)	*p*
Gender	Female	685 (79)	439 (83)	
	Male	185 (21)	91(17)	0.07
Age in years	<25	19 (2)	5 (1)	0.09
	25–34	261 (30)	90 (17)	<0.0001
	35–44	286 (33)	221 (42)	0.001
	45–54	215 (25)	157 (30)	0.04
	>55	89 (10)	57(11)	0.8
Time at workplace	1–5	228 (26)	81 (15)	<0.0001
	6–15	343 (39)	221 (42)	0.4
	16–25	215 (25)	168 (32)	0.002
	>26	84 (10)	11 (11)	0.3

HCW T1: Healthcare workers included at T1. HCW T2: Healthcare workers included at T2. *p*: *p* value.

**Table 4 ijerph-19-15260-t004:** Trends of work stressors’ impacts and psychosocial factor perceptions.

	Total Population			HCWs		
	T1	T2	*p* T1/T2	T1	T1	*p* T1/T2
	Mean (SD)	Mean (SD)		Mean (SD)	Mean (SD)	
Global work assessment	3.75 (0.81)	3.53 (0.8)	**<0.0001**	3.76 (0.78)	3.492 (0.77)	**<0.0001**
Workload	2.97 (0.7)	2,98 (0.7)	**0.03**	3.07 (0.7)	3.064 (0.64)	0.56
Abilities	3.04 (0.42)	3.01 (0.43)	0.16	3.02 (0.4)	2.992 (0.41)	0.2
Organizational context	2.83 (0.63)	2.79 (0.67)	0.16	2.76 (0.64)	2.7 (0.66)	0.06
Global work environment	3.7 (0.59)	3.19 (0.53)	**<0.0001**	3.7 (0.59)	3.14 (0.52)	**<0.0001**
Physical environment	3.9 (0.67)	3.8 (0.65)	0.68	3.9 (0.66)	3.91 (0.63)	0.9
Work demands	2.65 (0.66)	2.65 (0.63)	0.9	2.52 (0.6)	2.53 (0.6)	0.65
Global health	3.21 (0.66)	3.14 (0.67)	**0.017**	3.2 (0.66)	3.1 (0.65)	**0.01**
Physical health	3.52 (0.75)	3.4 (0.8)	**0.02**	3.5 (0.7)	3.35 (0.8)	**0.01**
Mental health	3.11 (0.83)	3.03 (0.8)	**0.03**	3.06 (0.85)	3 (0.81)	0.19
Stress	2.88 (0.82)	2.78 (0.8)	**0.004**	2.85 (0.82)	2.77 (0.8)	0.05
Physical symptoms	3.06 (1.2)	3.05 (1.15)	0.9	3.05(1.18)	3.01 (1.16)	0.56
Psychosomatic symptoms	3.47 (0.8)	3.39 (0.8)	**0.02**	3.49 (0.8)	3.38 (0.78)	**0.01**

T1: July–August 2020. T2: July–August 2021. Total population: Healthcare workers and non-healthcare subgroups. HCW: Healthcare worker subgroup. *p* T1/T2: *p* value of *t* test comparison between populations with questionnaires available (in 2020 T1 and 2021 T2).

## Data Availability

Data are available with request to the corresponding author.

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
