# Peer review of "Evolution of Global Health and Psychosocial Factors among Hospital Workers during First Year of SARS-CoV-2 Pandemic: A Longitudinal Study"

_ijerph, 2022, doi:10.3390/ijerph192215260_

Round 1

Reviewer 1 Report

Dear Authors

Thank you for the opportunity to review your manuscript reporting on a longitudinal study of the mental health of hospital workers during the SARS-CoV-2 pandemic.

Although there have been a number of surveys, there have been few cohorts where data were collected at more than one data point. The documentation of the negative impact of SARS-CoV-2 on the mental health of hospital workers, both health care workers and non-health care workers is valuable.

The Abstract describes the paper well.

Introduction: This section introduces a few studies, and then discusses guidelines and recommendations and returns to identify the need for further data. The introduction could highlight the strength of this paper as being a longitudinal study and the including of non-health workers by noting the paucity of such data in the current literature. The aim at the end of the introduction notes this intention but the reader has not been led through the introduction to the need for this type of study. Clarifying the need for this study would strengthen the paper.

It is possible that moving the comment about guidelines to the discussion would reduce the confusion that the paragraph p2 lines 60-63 creates as this currently interrupts the flow of the introduction.

Methods: The methods section requires a little more clarity. There is a strength by including health and non-health workers as this provides some comparison between different types of stress being experienced across the hospitals. In many hospitals non-health workers were still frontline workers and it is a little unclear if this is the case for the non-health workers here. Making a statement to explain this would be helpful.

The decision to email the participants through the occupational health service seems unusual. It is not clear if the participants had to have sought health care from that service to receive an email. If so, this could bias the data as it was gathered from people who determined they were ill at some stage and sought care. Also it is unclear though assumed that the email held a link to the survey and that the results did not go back to the occupational health service. This needs to be clearly stated to reassure the reader regarding the confidentiality of the survey. This is an important ethical issue that needs to be addressed.

It would be helpful to have access to more details about the SATIN questionnaire in a supplementary box. The details within the body of the paper about the 6 sections of this questionnaire are quite brief. It is unclear why an occupational physician would have 9 supplementary questions.

Similarly the scoring of the questionnaire using mean scores seems unusual and needs some justification to ensure this approach is validated, not just stating how the scores were interpreted but who these scores were derived/combined. The words ‘psychic health’ are unusual and perhaps mental health would be better here.

It appears that the results of the questionnaire at T1 and T2 could not be linked as the questionnaire was anonymous. If this is the case it should be explicitly stated as this would be helpful for the reader. However there is a suggestion in Table 2 that the number of people who participated in both T1 and T2 was know – and it is unclear how this could be if the questionnaire was truly anonymous. This detail is very important and needs to be explained. This is an important ethical issue that needs to be addressed.

There appears to have been an open question that gathered qualitative data yet this data is not presented in the results. It would be helpful to know why this was the case. (Is it to be presented elsewhere?; Has it been analysed?)

Results: While the tables of results are important to present, the current formatting of these tables seems difficult to follow. Moving across the table from T2 to T1 to the list of people in Table 1 is the reverse of the usual presentation of such data and the table should be revised.

Similarly, a revision of Table 2 and Table 3 would also assist the clarity of the presentation of these data for the reader who is confronted with lines of data and little orientation as to what these data mean.

There are comments under the tables that could be in the body of the results.

The results themselves need to have an articulate narrative that highlights the key points of the study. These will become the key points that can be further supported when compared to the current literature in the discussion. At the moment, the data are basically left to speak for themselves and the reader will not necessarily identify the key points that the authors wish to be the take-home messages from the numbers. Most of the current narrative seems to be re-stating the numbers in the table. The paper will be strengthened if the authors think about the key statements they would like to speak to the reader about.

I cannot comment on the statistical analyses themselves.  

The discussion offers some details from other studies, especially longitudinal studies which is helpful. However, these need to be integrated better with the findings of this study, highlighting how they differ or are similar and propose reasons for this. The finer details of the other studies that are reported in the discussion provide a complicated presentation and the actual key point that the authors wish to make is lost. Clear points in each section of the discussion would strengthen the presentation of the discussion. It is valuable for the authors to step the reader through the key issues that this study has identified and how they differ with or complement other studies. The reader can look up the reference if they wish to see the finer details of the study being referenced.

At present the key points get lost as the sentences lurch from one study to another and the linkage with the current study is often unclear. It is possible that as the key points emerging from the results have not been adequately defined, it has been difficult to then discuss these points in the context of the international literature. The discussion requires significant re-writing to enable the key message to be delivered in an effective manner.

In the discussion, the potential for bias has been reasonably well articulated.

It may be that the discussion is a better place for the authors to introduce the guidelines that they mention in the introduction but if this is the case then it needs to be adequately linked to the study itself.

The paper would also benefit from proof-reading again with a number of errors in sentence construction and missing words eg p2 line 56 and 93; p4 Table 2 (sexe and particiapnts); p6 line 177 and lines 203-206; p7 line 261; p 8 line 268 (non-ensuring) to highlight a few spots that require revision.

While the conclusion seems reasonable statements, the phrasing of the conclusion could be improved. The main issue with the conclusion is that it is difficult to identify how the data in this study supports these conclusions. The data did not seem to measure interventions per se. If the authors wish to highlight these issues in the conclusion, then the discussion would need to lead the reader on the journey towards these conclusions with more clarity.

I hope these comments are helpful. This paper has the potential to add to the literature but the results will need to be clarified and presented within the discussion more effectively to ensure this potential is reached.

Author Response

Responses to reviewer 1

dear reviewer,

thanks a lot for your relevant comments.

We try to respnd to eaxh point by point. If modifications done in the text linked to your comments, we put the new version. For some, we answered directly to your comments.

Hope you find them relevant

best regards

Dear Authors

Thank you for the opportunity to review your manuscript reporting on a longitudinal study of the mental health of hospital workers during the SARS-CoV-2 pandemic.

Although there have been a number of surveys, there have been few cohorts where data were collected at more than one data point. The documentation of the negative impact of SARS-CoV-2 on the mental health of hospital workers, both health care workers and non-health care workers is valuable.

The Abstract describes the paper well.

Introduction: This section introduces a few studies, and then discusses guidelines and recommendations and returns to identify the need for further data. The introduction could highlight the strength of this paper as being a longitudinal study and the including of non-health workers by noting the paucity of such data in the current literature.

L56 : For other hospital workers, not involved in healthcare, few studies have been published on SARS-CoV-2 pandemic impact on their mental and physical health. Most of them are Chinese and performed during the first outbreak. Comparing to administrative team, health professionals were 1.4 times more prone to feeling fear and suffering anxiety and depression (lu et al), and medical teams, especially those in Wuhan, had a higher level of stress than university students wu etal . Medical health workers showed higher prevalence rates of insomnia, anxiety, depression, somatization, and obsessive-compulsive symptoms than non-medical health workers (Zhang 2020). In his longitudinal study, Sasaki et al noted, after adjusting for the covariates, psychological distress (and subscales of fatigue, anxiety, and depression) as well as fear and worry of COVID-19 increased statistically significantly more among healthcare than non-healthcare workers from period 1 to period 2.

The aim at the end of the introduction notes this intention but the reader has not been led through the introduction to the need for this type of study. Clarifying the need for this study would strengthen the paper.

« In a previous study, we also found that being male and a HCW were risk factors for having a negative perception of work demand and the work activity environment (17).

l81: Out from France, HCWs had higher impact on their mental health at first wave of SARS-CoV-2 pandemic with higher deterioration during next waves. Psychosocial factors and working environment’ (physical and mental) assessment underlined occupational main causes of individual and collective working environment. In this field, SATIN is a transversal questionnaire developed for preventive medicine targeting well-being at work by the National Institute of Research on Security (INRS) in France (18, 19). Using this validated questionnaire, we performed a study on health and psychosocial factors in french hospital workers at the end of the first wave of  SARS-CoV-2 pandemic.

Indeed, recent narrative review points out the lack of studies and discussion about mechanisms that guarantees intervention and sustainability of them in literature on HCW mental health support program. For us the lack attention to sustainability may also be linked to focusing firstly disaster mental health rather than occupational health. (20).

So, it appears to us that better knowledge of physical and mental health impact on Hospital workers and occupational psychosocial factors with higher impact are clearly needed and relevant to enhance prevention program. « 

It is possible that moving the comment about guidelines to the discussion would reduce the confusion that the paragraph p2 lines 60-63 creates as this currently interrupts the flow of the introduction.

We did it and put it in discussion l 454

Methods: The methods section requires a little more clarity. There is a strength by including health and non-health workers as this provides some comparison between different types of stress being experienced across the hospitals. In many hospitals non-health workers were still frontline workers and it is a little unclear if this is the case for the non-health workers here. Making a statement to explain this would be helpful.

We rewrote and explain it :

l 126 : Medical residents, doctor, nurse, medical student, nurse assistant, midwife, paramedic, physiotherapist and radiographer were included in the subgroup “healthcare worker” (HCW), while other participants were included in the subgroup “non-healthcare worker” (non-HCW).  Administrative workers, laboratory technicians, maintenance, and cook staff workers were included non HWG subgroup. None of non HCW were frontline workers.

The decision to email the participants through the occupational health service seems unusual. It is not clear if the participants had to have sought health care from that service to receive an email. If so, this could bias the data as it was gathered from people who determined they were ill at some stage and sought care. Also it is unclear though assumed that the email held a link to the survey and that the results did not go back to the occupational health service. This needs to be clearly stated to reassure the reader regarding the confidentiality of the survey. This is an important ethical issue that needs to be addressed.

L 104 : « In 2020, during a hygiene and security committee, Hospitals’ trade unions, managers and occupational health service decided to assess Hospital workers global health. The Occupational Health Service built a project which was accepted by all members of the Hygiene and security committee. »

l114 : In 2021, through an e-mail from the Hygiene and security committee, we further invited all workers to participate to the second step of the study. The inclusion period was 01 July to 10 August 2021 (Time 2 (T2)).  The study was a questionnaire-based study using the same questionnaire in T1 and T2. In this second questionnaire, a specific question on participation in first step (T1) (yes/no) of our study was added. Participants are classified in three categories: P totT1 who only answered to questionnaire at T1, P tot T2 who only answered to the questionnaire at T2 and P tot T1andT2 workers who said yes at the specific question on first step T1 in the second questionnaire at T2.

It would be helpful to have access to more details about the SATIN questionnaire in a supplementary box. The details within the body of the paper about the 6 sections of this questionnaire are quite brief. It is unclear why an occupational physician would have 9 supplementary questions.

L148 :  Each question had five possible answers and each answer was linked to a specific score. Means of scores were calculated in each part of the questionnaire: health reports (physical health (self-evaluation of health and compared to next year), mental health (self-evaluation of mental health, confidence in the future), physical symptoms (musculoskeletal disorders), psychosomatic symptoms (headache, sleep problems, gastrointestinal problems), and stress (feeling stressed, exhausted at work, crack-up because of job), work strain (physical, emotional, concentration, knowledge), work abilities (physical, emotional, concentration, knowledge), working environment (physical environment,  work activity (interest, variety, utility, responsibility, diversity, quality of social relations), framework of activities (clarity, consistency, latitude, support, interruptions) and organizational context (number of hours, financial support, salary communication, job security, job  career), and self-assessment of work conditions in their entirety.

Similarly the scoring of the questionnaire using mean scores seems unusual and needs some justification to ensure this approach is validated, not just stating how the scores were interpreted but who these scores were derived/combined. The words ‘psychic health’ are unusual and perhaps mental health would be better here.

L 160 « For each question, workers had to choose only one in five possible answers and each answer is linked to a specific score. The scores obtained are continuous scores that can vary theoretically between 1 and 5 points; scores close to 1 indicate very poor health, while scores close to 5 indicate very good health on the dimension in question. Mean of scores were calculated in each part of the questionnaire: health reports (physical health, mental health,

It appears that the results of the questionnaire at T1 and T2 could not be linked as the questionnaire was anonymous. If this is the case it should be explicitly stated as this would be helpful for the reader. However there is a suggestion in Table 2 that the number of people who participated in both T1 and T2 was know – and it is unclear how this could be if the questionnaire was truly anonymous. This detail is very important and needs to be explained. This is an important ethical issue that needs to be addressed.

The questionanire was anonymous, but we put one specific question in the second questionanire at Time period 2 : do you particiate to first questionanire. In this view, we were able to select workers who answered yes in participating in both questionnaires and to compare thei answers on time 2 to the first survey at time 1 (all included population).

L 117 :  In this second questionnaire, a specific question on participation in first step (T1) (yes/no) of our study was added. Participants are classified in three categories: P totT1 who only answered to questionnaire at T1, P tot T2 who only answered to the questionnaire at T2 and P tot T1andT2 workers who said yes at the specific question on first step T1 in the second questionnaire at T2.

We also divided ech population between workers status: HCW or non HCW status

P HCW T1 are participants who answered only at first questionnaire (T1) and included in HCW group.

There appears to have been an open question that gathered qualitative data yet this data is not presented in the results. It would be helpful to know why this was the case. (Is it to be presented elsewhere?; Has it been analysed?)

  we did not present or analyse it in this paper.

Results: While the tables of results are important to present, the current formatting of these tables seems difficult to follow. Moving across the table from T2 to T1 to the list of people in Table 1 is the reverse of the usual presentation of such data and the table should be revised.

Similarly, a revision of Table 2 and Table 3 would also assist the clarity of the presentation of these data for the reader who is confronted with lines of data and little orientation as to what these data mean.

We revised all tables and apaologies for this mistake in formatting.

There are comments under the tables that could be in the body of the results.

We had some comments in the text but still not cut comments under for clarity of reading.

The results themselves need to have an articulate narrative that highlights the key points of the study. These will become the key points that can be further supported when compared to the current literature in the discussion. At the moment, the data are basically left to speak for themselves and the reader will not necessarily identify the key points that the authors wish to be the take-home messages from the numbers. Most of the current narrative seems to be re-stating the numbers in the table. The paper will be strengthened if the authors think about the key statements they would like to speak to the reader about.

For results of  different parts of SATIN questionnaire, when comparing mean scores of the total included population at T2 and T1, we found a significant deterioration of global health (p=0,006) and mental health (p=0,012) but not physical health. Workers reported higher negative perception of organizational context (p=0.03), global work assessment (p<0.0001), and also more psychosomatic symptoms (p=0.01) or self-evaluated stress (p=0.003).

Comparing the subgroup who completed the questionnaire at T1-T2 to the results of the first survey, we also found significant negative perception of global and mental health ( p<0.0001 and 0,005) but also workers at T2 reported more negative perception of organizational context (p <0.0001), global work assessment (p<0.0001), global health (p<0.0001), stress (p=0.002), work demand (p=0.001), global work environment (p=0.002).

I cannot comment on the statistical analyses themselves.  

The discussion offers some details from other studies, especially longitudinal studies which is helpful. However, these need to be integrated better with the findings of this study, highlighting how they differ or are similar and propose reasons for this. The finer details of the other studies that are reported in the discussion provide a complicated presentation and the actual key point that the authors wish to make is lost. Clear points in each section of the discussion would strengthen the presentation of the discussion. It is valuable for the authors to step the reader through the key issues that this study has identified and how they differ with or complement other studies. The reader can look up the reference if they wish to see the finer details of the study being referenced.

I agree with you, the key points were not clearly point in teh first version, we rewrote all the discussion with this idea in mind and hope it is now clearer and easier for the reader.

At present the key points get lost as the sentences lurch from one study to another and the linkage with the current study is often unclear. It is possible that as the key points emerging from the results have not been adequately defined, it has been difficult to then discuss these points in the context of the international literature. The discussion requires significant re-writing to enable the key message to be delivered in an effective manner.

In the discussion, the potential for bias has been reasonably well articulated.

It may be that the discussion is a better place for the authors to introduce the guidelines that they mention in the introduction but if this is the case then it needs to be adequately linked to the study itself.

L 466 : Indeed, guideline recommendations to reduce mental health burden in HCW during this pandemic was analyzed in a recent paper. They classified guidelines and recommendations in four categories including social/structural support, Work environment, communication/information and mental health support (21).

The paper would also benefit from proof-reading again with a number of errors in sentence construction and missing words eg p2 line 56 and 93; p4 Table 2 (sexe and particiapnts); p6 line 177 and lines 203-206; p7 line 261; p 8 line 268 (non-ensuring) to highlight a few spots that require revision.

We also made a specific revision on it

While the conclusion seems reasonable statements, the phrasing of the conclusion could be improved. The main issue with the conclusion is that it is difficult to identify how the data in this study supports these conclusions. The data did not seem to measure interventions per se. If the authors wish to highlight these issues in the conclusion, then the discussion would need to lead the reader on the journey towards these conclusions with more clarity.

We add such sentences at the beggining of the conclusion : We can hypothesize that main impact of SARS CoV-2 pandemic are on mental health and due to organizational context and work assessment deterioration. Occupational organization in hospital and especially in care units seems to be the most relevant axis of prevention on mental health impact.

I hope these comments are helpful. This paper has the potential to add to the literature but the results will need to be clarified and presented within the discussion more effectively to ensure this potential is reached.

Reviewer 2 Report

Please see the attached file with the comments and suggestions to improve your manuscript.

Author Response

Responses to reviewer 2

dear reviewer,

thanks a lot for your relevant comments.

We try to respnd to eaxh point by point. If modifications done in the text linked to your comments, we put the new version. For some, we answered directly to your comments.

Hope you find them relevant

best regards

we change the title

Title :

Evolution of global health and psychosocial factors among Hospital workers during first year of SARS-CoV- 2pandemic: a longitudinal study

INTRODUCTION

we add this part on non HCWs :

l 57 For other hospital workers, not involved in healthcare, few studies have been published on SARS-CoV-2 pandemic impact on their mental and physical health. Most of them are Chinese and performed during the first outbreak. Comparing to administrative team, health professionals were 1.4 times more prone to feeling fear and suffering anxiety and depression (lu et al), and medical teams, especially those in Wuhan, had a higher level of stress than university students wu etal . Medical health workers showed higher prevalence rates of insomnia, anxiety, depression, somatization, and obsessive-compulsive symptoms than non-medical health workers (Zhang 2020). In his longitudinal study, Sasaki et al noted, after adjusting for the covariates, psychological distress (and subscales of fatigue, anxiety, and depression) as well as fear and worry of COVID-19 increased statistically significantly more among healthcare than non-healthcare workers from period 1 to period 2.

For satin questionnaire in methods :

we only add demographics questions not used for statistics : The SATIN questionnaire was developed by University of Lorraine and the INRS (17). Based on the SATIN questionnaire, after authors acceptance, we add some questions on workplaces and demographics’ data. These added questions were not included in SATIN questionnaire results analysis but for demographic and work outcomes.

we develop the paragraph on it :

“Each question had five possible answers and each answer was linked to a specific score. Means of scores were calculated in each part of the questionnaire: health reports (physical health (self-evaluation of health and compared to next year), mental health (self-evaluation of mental health, confidence in the future), physical symptoms (musculoskeletal disorders), psychosomatic symptoms (headache, sleep problems, gastrointestinal problems), and stress (feeling stressed, exhausted at work, crack-up because of job), work strain (physical, emotional, concentration, knowledge), work abilities (physical, emotional, concentration, knowledge), working environment (physical environment,  work activity (interest, variety, utility, responsibility, diversity, quality of social relations), framework of activities (clarity, consistency, latitude, support, interruptions) and organizational context (number of hours, financial support, salary communication, job security, job  career), and self-assessment of work conditions in their entirety.

For each question, workers had to choose only one in five possible answers and each answer is linked to a specific score. The scores obtained are continuous scores that can vary theoretically between 1 and 5 points; scores close to 1 indicate very poor health, while scores close to 5 indicate very good health on the dimension in question. “

Statistics:

It is not clear in the first version. We only used means and standard deviations

“Data analysis was performed using R-4.0.2 and R studio software (R Foundation for Statistical Computing, Vienna, Austria). The results for continuous variables were shown as means . The ranked data, which were ranked from each part of the questionnaire, are presented as frequencies and percentages”

For the two groups:

“Medical residents, doctor, nurse, medical student, nurse assistant, midwife, paramedic, physiotherapist and radiographer were included in the subgroup “healthcare worker” (HCW), while other participants were included in the subgroup “non-healthcare worker” (non-HCW).  Administrative workers, laboratory technicians, maintenance, and cook staff workers were included non HWG subgroup. None of non HCW were frontline workers. “

we put these sentences in participants part as request l127.

We also clarify the psychosicial factors we assessed:

“health reports (physical health (self-evaluation of health and compared to next year), mental health (self-evaluation of mental health, confidence in the future), physical symptoms (musculoskeletal disorders), psychosomatic symptoms (headache, sleep problems, gastrointestinal problems), and stress (feeling stressed, exhausted at work, crack-up because of job), work strain (physical, emotional, concentration, knowledge), work abilities (physical, emotional, concentration, knowledge), working environment (physical environment,  work activity (interest, variety, utility, responsibility, diversity, quality of social relations), framework of activities (clarity, consistency, latitude, support, interruptions) and organizational context (number of hours, financial support, salary communication, job security, job  career), and self-assessment of work conditions in their entirety.”

For classifications:

“In first step of this study, in 2020, we divided demographic outcomes to perform multivariate analysis. A multinomial logistic regression analysis was performed and the associations between risk factors and outcomes after adjustment for cofounders including, gender, age, years at workplace. Demographic data were self-reported by the participants including occupation, gender (male/female), age (<35, 35-44, 45-54, >55 years), time at workplace (<5, 6-15, 16-26,>26 years).

Refering to this first publication and to allow better reliability between results of the two phases we decide to keep such classification by categories. “

results:

we add in methodology for demographic variables:

In first step of this study, in 2020, we divided demographic outcomes to perform multivariate analysis. A multinomial logistic regression analysis was performed and the associations between risk factors and outcomes after adjustment for cofounders including, gender, age, years at workplace. Demographic data were self-reported by the participants including occupation, gender (male/female), age (<35, 35-44, 45-54, >55 years), time at workplace (<5, 6-15, 16-26,>26 years).

We rewrote all tables and apologies for bad formatting of it.

Participants:

 l196:

“A total of 1407 participants completed the questionnaire at T1 and 835 at T2. 568 workers completed the questionnaire at T1 and T2. In the subgroup of HCW, available for analysis. In the subgroup of non HCW, respectively for T1, T2 and T1-T2, 476, 304 and 211 questionnaires were included. Due to time at hospital less than one year we exclude 344 workers in the first step and 4 in the second step.

Most of HCWs were nurses (26,4 and 36,3%), nurses assistants (13,4 and 6,5%) and medical doctors (10 and 11,6%) for the two periods. Mots of included workers are female (79,2 and 82,4% and HCWs (66,2 and 64%). In the non HCWs’ group, administrative assistant  (10 and 11,6%), technicians,and laboratory(3,1 and 2,9%), and technicians (1,8 and 1,2%) are most represented. Most of workers were aged between 35 and 44 years old.”

We only use gender in all the paper and tables

after reading your comments on rationale of comparison, we follow your suggestions and compare results at T1 results from subjects who completed both assessments.

In Table 4 yes it is means and SD and corrections in methods was made.

For table 2 and the 300 participants did not meet criterion, they were excluded: l 199

“Due to time at hospital less than one year we exclude 344 workers in the first step and 4 in the second step. “

we rewrote the discussion.

We consider not including non HCWs but for other reviewer and us, this is one of study’ strenghts.

Round 2

Reviewer 2 Report

The authors improve many sections of the manuscript. However, I have still major concerns regarding the statistical procedure and the results section. Both can be improved and make a better manuscript. 

Table 1. The numbers are not consistent. In the text, the authors state that 1407 participants were included, However when the sum of several other variables was performed, the number is 1405. Also, the 334 subjects that had less than one year in the workplace were not excluded.  In addition, the total number of subjects in this variable is 1507, therefore, if the 334 participants that should be excluded (due to exclusion criteria) were correctly removed, the number of participants should be 1236. These subjects should be removed since the beginning of the analysis due to the exclusion criteria (as previously mentioned) For T1. The same thing is presented for T2 in Table 1. 830 participants were registered without removing the 4 subjects who reported having worked there for less than a year and also in the text it is mentioned that there were835 participants at T2 (829 for the variable work status.

Also the variable Time at Workplace has to be arranged as the order of the ordinal categories is not in order as in age (ascendant).

Table 2. Comparisons should be made for each variable and not for the values obtained. For example, one analysis for gender, one analysis for age which do not appear in the table. Also, the subjects with less than one year at workplace were not excluded.

It seems that the comparisons were between the complete sample who answered T1 and then compared it with the sample that answered T1 and T2. The comparison should be made only for those who completed T1 and T2 using repeated samples t test for continuous variables and McNemar tests for categorical variables and Wilcoxon tests for the age and time at workplace analyses (ordinal variables), as these are not independent groups. The same should be used for the exclusive comparison of the HCW.

For example, all the tables should have the format of Table 1 where T1 and T2 data are presented and then the comparison between them (with the same number of subjects at both evaluations). There is no need to compare “results from workers who completed both 109 assessments and those who only have first assessment.” As it is not the aim of the study and only provides biased information for a follow-up study.

Please, verify the statistical strategy you are using to give clearer results for this important follow-up work. According to the results obtained, adequate the discussion section.

Author Response

Dear Reviewer,

thanks for your comments.

For more clarity, we compare two populations : the total population who answered to the questionnaire at T1 and the total population who answered at T2.

Maybe we lost some relevance, but agree with you it is a clearest way.

W also talked about statistical strategy with colleagues and finalize it as described in the paper.

« Data analysis was performed using R-4.0.2 and R studio software (R Foundation for Statistical Computing, Vienna, Austria). The results for continuous variables were shown as means. The ranked data, which were ranked from each part of the questionnaire, are presented as frequencies and percentages. The mean change in scores of psychological distress and physical symptoms from T1 to T2 were compared between all workers and in the subgroup of HCW (t-test for two independent groups). Chi 2 test was used for comparison of demographic data between the two populations. The significant level was set at p = 0.05. »

The questionnaire is anonymous and it is not possible to specify the population who those who completed T1 and T2.

We also check all data, exclude before analysis the workers with less than one year of experience.

We arrange table 1 with Time at Workplace as the order of the ordinal categories.

In our questionnaire variables including age and time at workplace are only in class and we are not able to make other comparison than catégorials.

We rewrote the discussion in regard of new data.

Hope it is convenient to your expertise.
